# Competitive repopulation of an empty microglial niche yields functionally distinct subsets of microglia-like cells

Harald Lund [1], Melanie Pieber [1], Roham Parsa [1], Jinming Han[1], David Grommisch[1], Ewoud Ewing [2], Lara Kular[2], Maria Needhamsen[2], Alexander Espinosa[3], Emma Nilsson[4], Anna K. Överby[4], Oleg Butovsky[5,6], Maja Jagodic[2], Xing-Mei Zhang[1] & Robert A. Harris [1]

Circulating monocytes can compete for virtually any tissue macrophage niche and become long-lived replacements that are phenotypically indistinguishable from their embryonic counterparts. As the factors regulating this process are incompletely understood, we studied niche competition in the brain by depleting microglia with >95% efficiency using $Cx3cr1^{CreER/+}R26^{DTA/+}$ mice and monitored long-term repopulation. Here we show that the microglial niche is repopulated within weeks by a combination of local proliferation of $CX3CR1^+F4/80^{low}Clec12a^-$ microglia and infiltration of $CX3CR1^+F4/80^{hi}Clec12a^+$ macrophages that arise directly from $Ly6C^{hi}$ monocytes. This colonization is independent of blood brain barrier breakdown, paralleled by vascular activation, and regulated by type I interferon. $Ly6C^{hi}$ monocytes upregulate microglia gene expression and adopt microglia DNA methylation signatures, but retain a distinct gene signature from proliferating microglia, displaying altered surface marker expression, phagocytic capacity and cytokine production. Our results demonstrate that monocytes are imprinted by the CNS microenvironment but remain transcriptionally, epigenetically and functionally distinct.

[1] Applied Immunology and Immunotherapy, Department of Clinical Neuroscience, Karolinska Institutet, Center for Molecular Medicine, Karolinska Hospital Solna, Stockholm 17176, Sweden. [2] Department of Clinical Neuroscience, Karolinska Institutet, Center for Molecular Medicine, Karolinska Hospital Solna, Stockholm 17176, Sweden. [3] Unit of Rheumatology, Department of Medicine, Karolinska Institutet, Center for Molecular Medicine, Karolinska Hospital Solna, Stockholm 17176, Sweden. [4] Department of Clinical Microbiology, Virology, Umeå University, Umeå 90185, Sweden. [5] Ann Romney Center for Neurologic Diseases, Department of Neurology, Brigham and Women's Hospital, Harvard Medical School, Boston 02115 MA, USA. [6] Evergrande Center for Immunologic Diseases, Brigham and Women's Hospital, Harvard Medical School, Boston 02115 MA, USA. Correspondence and requests for materials should be addressed to R.A.H. (email: Robert.Harris@ki.se)

All organs in the body house resident macrophage populations with duties that are tailored to the function of that organ. For example, brain microglia promote neuronal wiring, splenic red pulp macrophages (RPM) eliminate dying erythrocytes in the spleen and alveolar macrophages recycle surfactant in the lung. It is now well-recognized that the majority of these tissue macrophages are derived from embryonic precursors that self-maintain throughout adulthood[1–4] rather than from continuous monocyte input, as was previously suggested[5]. During development of the embryo, tissues are seeded with successive waves of hematopoietic precursors from the yolk sac and fetal liver[1,6–8], which following birth are displaced by BM-derived monocytes in some organs[9–14]. Furthermore, upon infection or injury, circulating blood monocytes can give rise to long-lived replacements of macrophages in many organs including the heart[15], liver[16], lung[17], and brain[18,19]. The final macrophage composition can thus be heterogeneous across organs, and this has complicated recent attempts to assign macrophage nomenclature based on their ontogeny[20]. Whether the ontogeny of resident macrophage populations ultimately dictates functionality is not well understood and may vary depending on the circumstances generating the macrophage[21].

The diversity of tissue macrophage populations has been proposed to be the result of exposure to specialized microenvironments[8,22–25]. Significant reprogramming of macrophage precursors transplanted into adult tissue microenvironments has been demonstrated[24,26,27], arguing for an imprinting capacity of the macrophage niche as was recently proposed[28]. Peripheral myeloid cells that colonize the microglial niche adopt microglia-like morphology[29,30], display ATP-sensing capacity[29], can promote repair following cranial irradiation[19] and require TGF-β signaling for functional integration into the central nervous system (CNS)[31]. However, to what degree monocytes become imprinted by the microglial niche and what factors regulate this imprinting remain unknown.

In this study, we develop a novel strategy to efficiently deplete microglia, and observe competitive repopulation giving rise to a permanent mosaic of myeloid cells derived both from proliferating microglia and from brain-engrafting Ly6C$^{hi}$ monocytes. We provide evidence of niche imprinting, as monocyte-derived macrophages adopt both microglia gene expression and epigenetic profiles. However, monocyte-derived macrophages display a unique gene signature, giving rise to a distinct surface marker phenotype and functional profile. Our results demonstrate that both niche imprinting and myeloid origin define macrophage identity within the CNS, and this may have implications for understanding macrophage biology, disease, and therapy.

## Results

**Microglial depletion-repopulation in $Cx3cr1^{CreER/+}$ $R26^{DTA/+}$.** In order to make the CNS myeloid niche available we aimed to efficiently deplete microglia. To achieve both fast and efficient depletion we elected to breed mice with tamoxifen (TAM)-inducible Cre under the microglia-expressed CX3CR1-promoter ($Cx3cr1^{CreER-EYFP}$ mice)[32] with $R26^{DTA/+}$ mice[33], which results in intracellular diphtheria toxin A (DTA) expression upon Cre recombination. TAM administration to $Cx3r1^{CreER/+}R26^{DTA/+}$ mice resulted in ~95% loss of CD11b$^+$CD45$^+$Ly6C$^-$Ly6G$^-$ CX3CR1$^+$ microglia by day 7 (Fig. 1a, b, gating strategy Supplementary Fig. 1). We confirmed this high efficiency of depletion by immunohistochemistry using microglia-specific[34] P2ry12 and CX3CR1-YFP antibodies (Fig. 1d), as well as by loss of expression of the key microglial transcripts *P2ry12*, *Cx3cr1*, and *Siglech* (Supplementary Fig. 2a).

We analyzed additional groups of mice at day 28, and as expected from previous reports,[29,30,35] we detected repopulation of the CX3CR1$^+$ compartment. Flow cytometric analyses revealed two distinct populations within the CX3CR1$^+$ gate that differentially expressed the F4/80 antigen (Fig. 1a), and this was confirmed by immunohistochemistry (Supplementary Fig. 2B). The surface phenotype of the CX3CR1$^+$F4/80$^{low}$ population overlapped with microglia from control mice, whereas the CX3CR1$^+$F4/80$^{hi}$ population expressed higher levels of CD45 and lower levels of Siglec H (Fig. 1c). Siglec H has previously been suggested to be specifically expressed by microglia[36]. Importantly, the appearance of the F4/80$^{hi}$ population was not transient, since they remained 12 weeks after TAM administration (Supplementary Fig. 2D) and retained their surface marker phenotype (Supplementary Fig. 2E). In further support of repopulation by two distinct types of microglia we observed that the homogenous distribution of CX3CR1$^+$P2ry12$^+$ microglia in control brains was replaced by day 28 with pockets of CX3CR1$^+$P2ry12$^+$ and CX3CR1$^+$P2ry12$^-$ microglia-like cells (Fig. 1d). The P2ry12 antibody has been proposed to only label microglia and not peripherally-derived myeloid cells[34] and we observed that F4/80$^{hi}$ parenchymal macrophages were indeed P2ry12$^-$ (Supplementary Fig. 2C). qPCR analysis of whole brain further confirmed gradual increases in *P2ry12*, *Cx3cr1* and *Siglech* expression from days 7 to 28 (Supplementary Fig. 2A).

**Local proliferation and infiltration repopulate the niche.** Based on these data we hypothesized that the F4/80$^{low}$ cells represented CNS-resident microglia that had expanded following depletion and that F4/80$^{hi}$ cells were peripherally derived. To more conclusively address the origin of CNS-repopulating F4/80$^{low}$ and F4/80$^{hi}$ microglial cells we irradiated $Cx3cr1^{CreER/+}R26^{DTA/+}$ and control $Cx3cr1^{CreER/+}$ mice (CD45.2) and reconstituted them with congenic (CD45.1) BM. To limit the effects of irradiation on the ability of microglia to repopulate the niche[30,37] we protected the head from irradiation. After 8 weeks of reconstitution we administered TAM and analyzed the chimeras after a further 8 weeks. Head-protected CD45.1 → $Cx3cr1^{CreER/+}R26^{DTA/+}$ chimeras displayed frequencies of F4/80$^{low}$ and F4/80$^{hi}$ cells that were similar to those in non-irradiated mice (Fig. 2a). Using this system we could thus demonstrate that F4/80$^{low}$ cells were completely host (CD45.2)-derived and that F4/80$^{hi}$ cells were completely donor (CD45.1)-derived (Fig. 2a). To control for irradiation-induced damage we used a chemotherapy regimen to achieve myeloablation, a strategy that has previously been reported to lead to almost complete donor chimerism without spontaneously inducing CNS myeloid engraftment[38,39]. We thus produced $Cx3cr1^{GFP/+}Ccr2^{RFP/+}$ → $Cx3cr1^{CreER/+}R26^{DTA/+}$ chimeras where donor and host CX3CR1$^+$ cells could be differentiated by GFP and YFP expression, respectively (Supplementary Fig. 3). Using this experimental setup we found that after depletion of microglia, practically all F4/80$^{hi}$ cells were GFP$^+$ (Fig. 2b), confirming their peripheral origin.

These results supported our hypothesis of a dual origin of CNS repopulating macrophages and further suggested the involvement of a combination of local microglial proliferation and infiltration of BM-derived precursors in order to repopulate the niche. This was supported by high frequencies of Ki67$^+$ microglia at day 7 and their absence at day 28 (Fig. 2c). To conclusively address whether the F4/80$^{low}$ population arose by microglial proliferation in response to depletion we administered EdU during the repopulation phase, which confirmed a burst of proliferation in F4/80$^{low}$ microglia during days 0–14 (Fig. 2d).

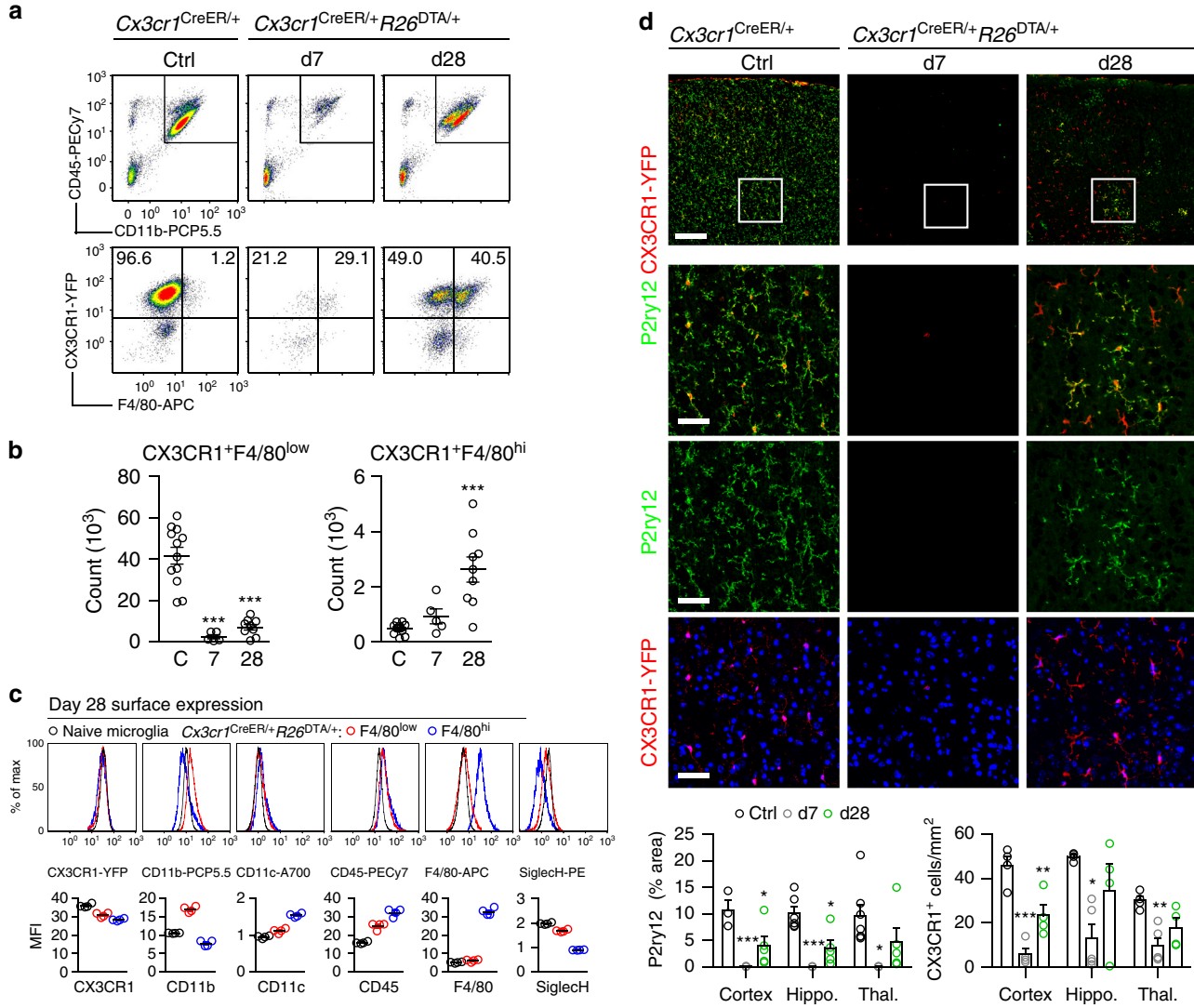

**Fig. 1** Kinetics of depletion and repopulation of microglia in $Cx3cr1^{CreER/+}R26^{DTA/+}$ mice. **a** Dot plots detailing depletion and repopulation of microglia in $Cx3cr1^{CreER/+}R26^{DTA/+}$ mice on the indicated days after TAM administration. Top panel is gated on live singlet cells. Bottom panel is gated on CD11b$^+$CD45$^+$Ly6C$^-$Ly6G$^-$. **b** Quantification of CX3CR1$^+$F4/80$^{low}$ and CX3CR1$^+$F4/80$^{hi}$ populations by flow cytometry. $n = 12, 5, 9$ mice, mean ± s.e.m. ***$p < 0.001$ by one-way ANOVA with Dunnett's Multiple Comparison test. Data are pooled from three separate experiments. The experiment was repeated twice for each time point. **c** Surface expression by flow cytometry on repopulated F4/80$^{low}$ and F4/80$^{hi}$ cells at day 28 compared to naive microglia. Gated on CD11b$^+$CD45$^+$Ly6C$^-$Ly6G$^-$. $n = 4$ mice/group. Lines represent mean values. The experiment was performed twice. **d** Quantification of P2ry12 and CX3CR1-YFP staining in the indicated brain regions. Representative images are from the cortex. $n = 3$–$7$ mice/group, mean ± s.e.m. ***$p < 0.001$, **$p < 0.01$, *$p < 0.05$ by one-way ANOVA with Dunnett's Multiple Comparison test. Scale bar top panel 200 μm, bottom three panels 50 μm. Hippo. Hippocampus. Thal. Thalamus

**BM-derived microglial replacements are Ly6C$^{hi}$ monocytes.** While Ly6C$^{hi}$ monocytes can act as precursors of many peripheral macrophage populations[21], BM-derived microglia have been proposed to arise from myeloid progenitors without passing through a monocyte intermediate[30,40]. Furthermore, it has been suggested that blood brain barrier (BBB) disruption is a prerequisite for brain colonization by peripheral myeloid cells[41]. Our experiments using head protection and chemotherapy-induced myeloablation demonstrated that brain irradiation was not required for niche competition by peripheral precursors. In addition, we did not detect disruption of the BBB as a consequence of microglial depletion (Supplementary Fig. 4A). Instead, we recorded increased mRNA levels in the CNS of the chemokine receptor *Ccr2* and of the monocyte chemoattractants *Ccl2, Ccl3, Ccl4,* and *Ccl5* at day 7 (Supplementary Fig. 4B), suggesting chemotactic recruitment of Ly6C$^{hi}$ monocytes. In addition, we detected elevated mRNA levels of several other

cytokines and chemokines at day 7, but most were back to baseline by day 14 (Supplementary Fig. 4C). This was accompanied by astrocytosis, as evidenced by both increased *Gfap* mRNA expression (Supplementary Fig. 4D) and GFAP immunostaining (Supplementary Fig. 2E).

To address the importance of Ly6C$^{hi}$ monocyte release from the BM we produced WT:$Ccr2^{-/-}$ → $Cx3cr1^{CreER/+}R26^{DTA/+}$ mixed BM chimeras (Fig. 3a) that lack CCR2$^{-/-}$ monocytes in the circulation (ref. [42] and Fig. 3b). We observed that the CNS F4/80$^{hi}$ compartment was exclusively repopulated by WT cells (Fig. 3a), demonstrating a crucial role of CCR2 in this process. Consistent with this observation, a time course experiment encompassing depletion/repopulation revealed a wave of Ly6C$^{hi}$ monocytes entering the brain that peaked at day 2 after TAM, prior to the establishment of the F4/80$^{hi}$ macrophage pool (Fig. 3b). At the peak of monocyte infiltration into the CNS we observed vascular activation in the brain, as evidenced by

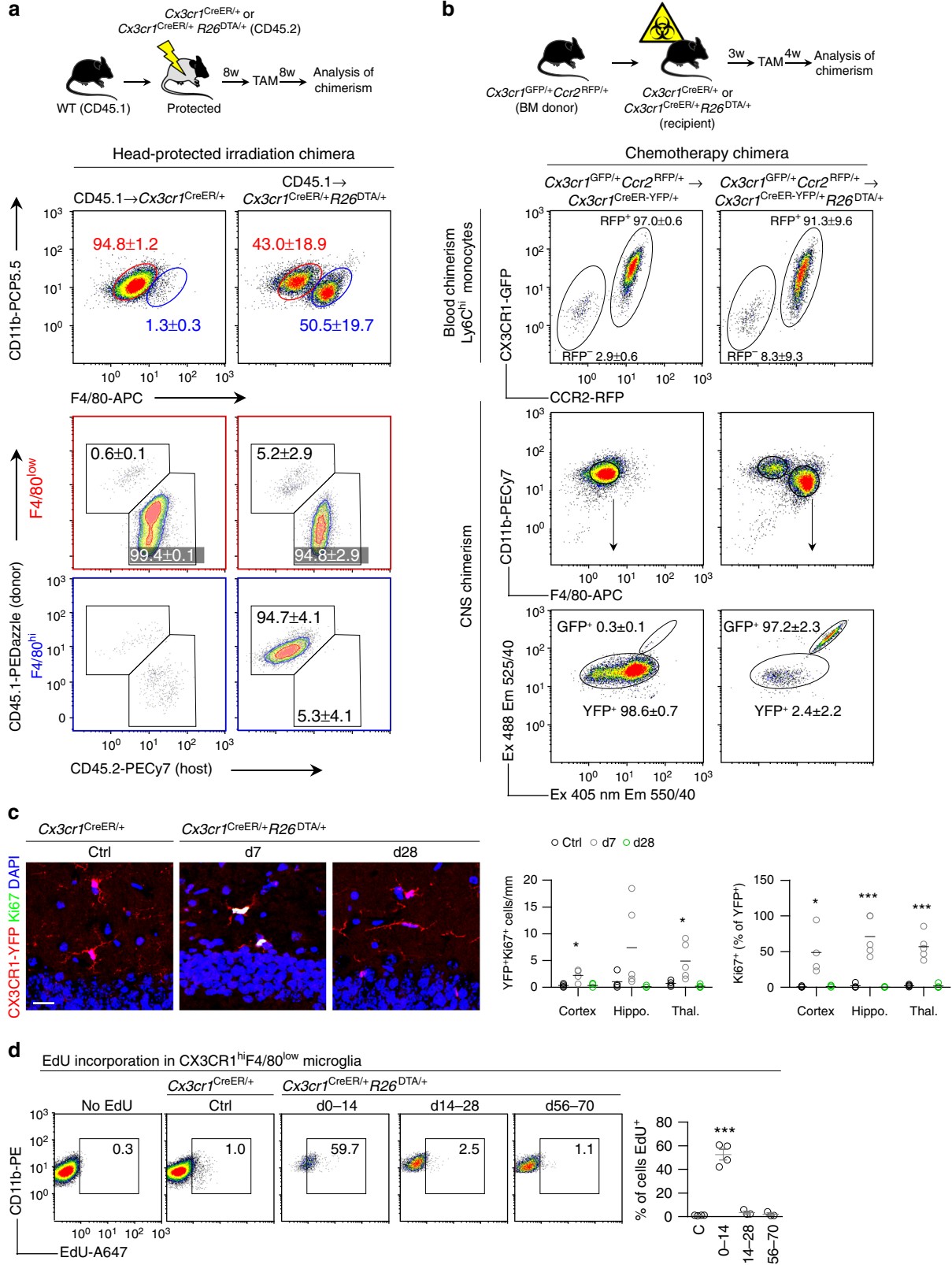

**c** Ctrl | d7 | d28 | ○ Ctrl | ○ d7 | ○ d28

**d** EdU incorporation in CX3CR1^hi^F4/80^low^ microglia

increased ICAM-1 staining (Fig. 3c). This suggested that peripherally derived macrophages could enter the CNS through the vasculature. Furthermore, Iba-1 staining did not reveal obvious infiltrates in meninges or choroid plexus, arguing against these anatomical areas as major infiltration sites (Supplementary Fig. 5A).

CCR2 is not an exclusive marker for Ly6C^hi^ monocytes and can also be expressed on myeloid precursors and stem cells. However, we did not detect increased numbers of c-kit^+^ progenitor cells in either the CNS (Supplementary Fig. 5B) or the blood (Supplementary Fig. 5C), arguing against repopulation from BM-derived stem cell/progenitors[43]. Furthermore, we did

**Fig. 2** Repopulating microglia have dual origins. **a** Analysis of chimerism in F4/80$^{low}$ and F4/80$^{hi}$ populations in CD45.1 chimeras. Top panel gated on CD11b$^+$F4/80$^+$Ly6C$^-$Ly6G$^-$. Bottom panels gated on F4/80$^{low}$ and F4/80$^{hi}$ as indicated. Percentages are mean ± s.d. of $n = 5$ mice/group. The experiment was performed twice. **b** Analysis of chimerism in Cx3cr1$^{GFP/+}$Ccr2$^{RFP/+}$ → Cx3cr1$^{CreER/+}$R26$^{DTA/+}$ chimeras that had received busulfan chemotherapeutic for myeloablation. Successful separation of GFP$^+$ and YFP$^+$ populations is demonstrated in Supplementary Fig. 3. Blood is gated on CD11b$^+$Ly6G$^-$CD115$^+$Ly6C$^{hi}$ and CNS is gated on CD11b + Ly6C$^-$. Values in plots are mean ± s.d of $n = 4$ and six mice. The experiment was performed once. **c** Quantification of CX3CR1-YFP$^+$Ki67$^+$ proliferating microglia in the indicated brain regions. Representative images are from the hippocampus. $n = 4$–5 mice/group. Lines represent mean values. ***$p < 0.001$, *$p < 0.05$ by one-way ANOVA with Dunnett's Multiple Comparison test. Scale bar 20 μm. **d** Proliferation in repopulating F4/80$^{low}$ microglia assessed by EdU. EdU was administered in the drinking water for 14 days during the indicated time periods after TAM. Control mice were given EdU days 0–14 after TAM. Gated on CD11b$^+$Ly6C$^-$Ly6G$^-$CX3CR1$^+$F4/80$^{low}$. $n = 3$–4 mice/group, the experiment was performed twice. Lines represent mean values. ***$p < 0.001$ by one-way ANOVA with Dunnett's Multiple Comparison test

not detect changes in neutrophils or lymphocyte numbers in the CNS following depletion (Supplementary Fig. 5D-E). More importantly however, adoptive transfer of purified Ly6C$^{hi}$ monocytes from Cx3cr1$^{GFP/+}$Ccr2$^{RFP/+}$ mice into microglia-depleted Cx3cr1$^{CreER/+}$R26$^{DTA/+}$ mice during the peak of monocyte entry into the brain (days 0, 1, and 2) resulted in specific reconstitution of >70% of the F4/80$^{hi}$ compartment without contributing to the F4/80$^{low}$ microglial pool (Fig. 3d). Collectively, our results suggested a sequence of events whereby the remaining microglia proliferate extensively in order to refill the niche following depletion. Monocytes simultaneously infiltrate the brain and give rise to F4/80$^{hi}$ macrophages.

To address whether Ly6C$^{hi}$ monocytes enter in one wave or continuously infiltrate the brain, we administered EdU at different time points following depletion. CNS-retrieved Ly6C$^{hi}$ monocytes were consistently ~80% EdU$^+$. When EdU was given during days 0–14 after TAM, the F4/80$^{hi}$ macrophages were also ~80% EdU$^+$. However, when EdU is administered during days 14–28 or days 56–80 after TAM, EdU incorporation into the F4/80$^{hi}$ pool was not different from controls (Supplementary Fig. 5F). This indicates that F4/80$^{hi}$ macrophages are generated from Ly6C$^{hi}$ monocytes entering the brain directly following depletion.

Taken together, our data imply that in Cx3cr1$^{CreER/+}$R26$^{DTA/+}$ mice the availability of an empty niche allows circulating monocytes to infiltrate the brain and to give rise to long-lived microglia-like cells.

**Engrafting macrophages adopt TGF-β driven transcription.** Cx3cr1$^{CreER/+}$R26$^{DTA/+}$ mice thus represent a useful tool to study the impact of the CNS microenvironment on peripherally-derived macrophages. To investigate whether monocyte-derived macrophages adopted microglia gene expression in the microglial niche, we depleted microglia and profiled the transcriptomes of sorted F4/80$^{low}$ and F4/80$^{hi}$ populations (Fig. 4a) after 4 and 12 weeks. In parallel we sorted Ly6C$^{hi}$ monocytes, common myeloid progenitors (CMP) and granulocyte-macrophage progenitors (GMP) from the BM, as well as splenic RPM and intestinal macrophages from naive mice for comparison. Hierarchical clustering of the 1000 most variable genes in the dataset separated samples according to tissue origin (Supplementary Fig. 6). Principal component analysis (PCA) based on all genes similarly organized all CNS macrophages into one cluster (Fig. 4b) including F4/80$^{low}$ and F4/80$^{hi}$ subsets as well as naive microglia and recently proliferated microglia sorted from Cx3cr1$^{CreER/+}$R26$^{DTR/+}$ mice, which express the diphtheria toxin receptor (DTR)[30,32]. This indicated a closer relationship between F4/80$^{hi}$ macrophages and microglia compared to their BM precursors or other tissue macrophages, which clustered separately. This was further substantiated by the fact that F4/80$^{hi}$ macrophages expressed several genes that have previously been described to be microglia-specific[44,45] (Fig. 4c), including Fcrls, P2ry12, P2ry13, Siglech, Olfml3, and Tmem119. In addition, several transcription factors that are associated with microglia

homeostasis[46] were detected in F4/80$^{hi}$ macrophages, including Egr1, Mafb, Mef2a, and Jun, with the exception of Sall1 (ref.[37]) (Fig. 4d). Several of these genes are induced by TGF-β signaling[34] and we have recently demonstrated that abrogation of TGF-β signaling in monocyte-derived macrophages results in fatal demyelinating disease[31]. Consistently, we observed high levels of Tgfbr1 expression in both F4/80$^{low}$ and F4/80$^{hi}$ populations, which were similar to levels in naive microglia (Fig. 4e). These results indicate that monocytes adopted key microglia-specific genes subsequent to CNS engraftment.

**Repopulating cells differ transcriptionally and functionally.** Comparison of the transcriptomes of naive microglia with other populations resulted in 4967 differentially expressed genes (>2 fold both directions, adj. $p < 0.05$) with Ly6C$^{hi}$ monocytes, 5240 with RPMs and 4370 with intestinal macrophages, respectively. F4/80$^{low}$ microglia 4 weeks after TAM administration differentially expressed 361 genes compared to naive microglia, and only 56 genes differed after 12 weeks (Fig. 5a). For F4/80$^{hi}$ macrophages the corresponding numbers were 886 and 1173, respectively (Fig. 5a). This indicated that after repopulation the F4/80$^{low}$ transcriptome is affected, but almost returns to baseline within 12 weeks. Conversely, F4/80$^{hi}$ macrophages do not completely adopt the naive microglia transcriptome. This was further supported by visualization of the genes that were differentially expressed in F4/80$^{hi}$ macrophages compared to naive microglia (Fig. 5b).

To further control for the inflammatory environment occurring as a result of depletion we also compared the transcriptomes of repopulated F4/80$^{low}$ and F4/80$^{hi}$ macrophages at 12 weeks directly, these cells being sorted from the same brains and thus exposed to the same cytokine environment. This analysis revealed that of the 850 genes differentially expressed between F4/80$^{low}$ and F4/80$^{hi}$ macrophages, 769 were also differentially expressed between F4/80$^{hi}$ macrophages and naive microglia (Fig. 5c). Among the genes most importantly downregulated in F4/80$^{hi}$ macrophages compared to both naive and repopulated F4/80$^{low}$ microglia (Fig. 5d), we identified two Spalt-like transcription factors (Sall3 and Sall1), confirming previous reports of their specific expression in microglia[24,25,34,37]. Furthermore, Upk1b, Slc2a5, St3gal5, Jam2, Adgrg1 were other genes that were highly downregulated in F4/80$^{hi}$ macrophages (Fig. 5d). Among the genes most prominently upregulated in F4/80$^{hi}$ macrophages compared to both naive and repopulated F4/80$^{low}$ microglia (Fig. 5d), we identified C-type lectins (Clec12a, Clec4n, Mrc1) and scavenger receptors (Msr1, Cd36). Furthermore, we observed high expression of cell adhesion molecules associated with transendothelial migration (Itga4, Vcam1, Itgal), which concords with the vascular activation observed in microglia-depleted brains (Fig. 3c). Furthermore, transcription factors normally associated with monocyte development and monocyte-derived tissue macrophages (Ahr, Tfec, Runx3, Spic)[24,47,48] were more highly

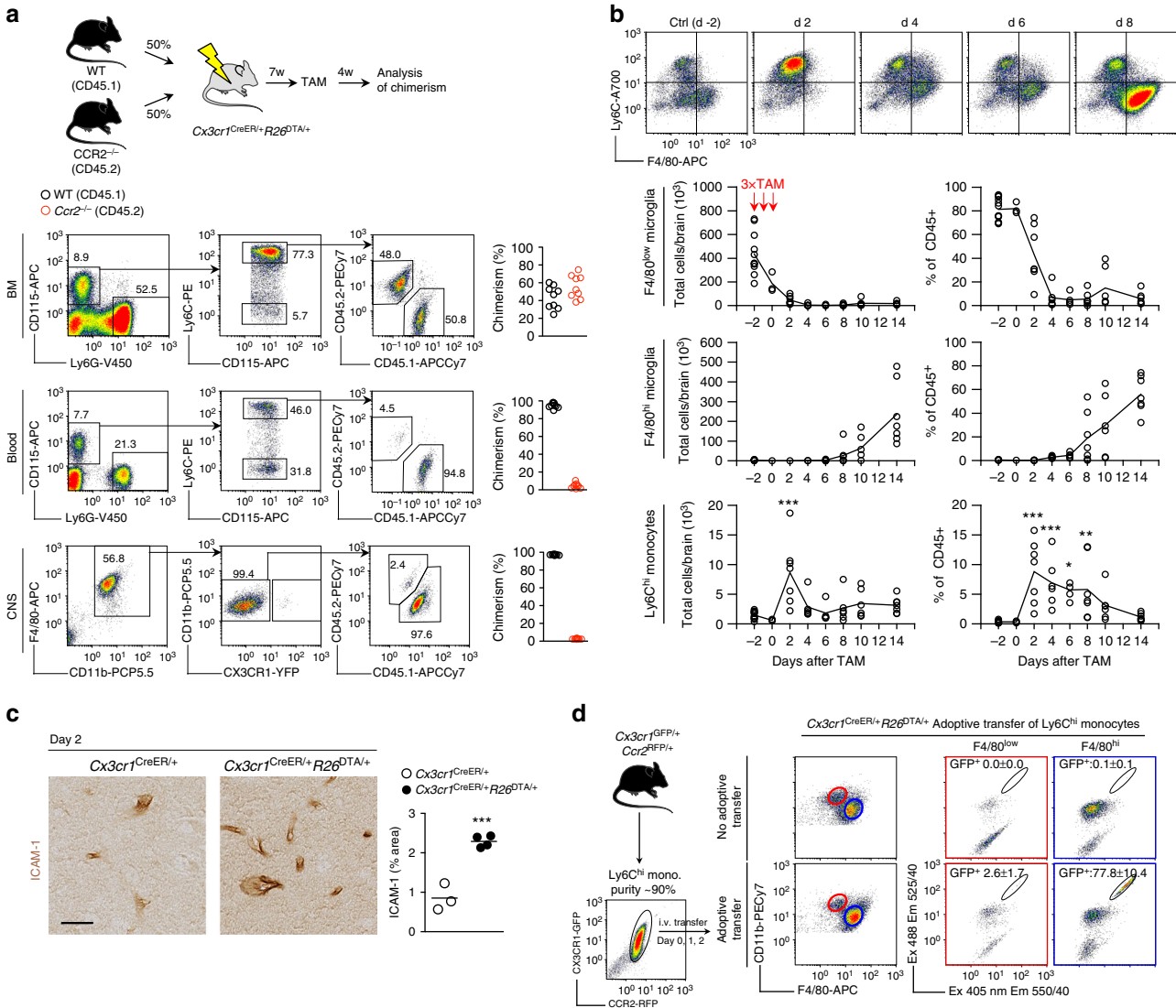

**Fig. 3** F4/80[hi] macrophages are derived from CCR2[+]Ly6C[hi] monocytes. **a** Analysis of BM, blood and CNS chimerism in WT:Ccr2[−/−] → Cx3cr1[CreER/+]R26[DTA/+] competitive chimeras. In all organs gated on live singlet cells. n = 9 mice. The experiment was performed twice. **b** Kinetic analysis of Ly6C[hi] monocytes in relation to F4/80[low] microglia and F4/80[hi] macrophages in the CNS during depletion in Cx3cr1[CreER/+]R26[DTA/+] mice. Representative flow cytometry plots are gated on CD11b[+]CD45[hi]Ly6G[−] cells. Ly6C[hi], F4/80[low], and F4/80[hi] gated as indicated in Supplementary Fig. 1. Data is from two pooled experiments. n = 12, 3, 7, 6, 5, 7, 6, 7 mice/time point. Connecting lines represent mean values. **p < 0.01, ***p < 0.001 by one-way ANOVA with Dunnett's Multiple Comparison test. **c** Activation of the cerebral vasculature as assessed by ICAM-1 staining on day 2 after TAM administration. Quantification was performed in the thalamus on n = 3 and 4 mice/group and 3 sections/mouse. Scale bar 25 μm. Lines represent mean values. ***p < 0.001 by Student's unpaired two-tailed t-test. **d** Analysis of reconstitution of the F4/80[low] and F4/80[hi] compartments after adoptive transfer of 1.7–3.0 × 10[6] Ly6C[hi] monocytes/day from Cx3cr1[GFP/+]Ccr2[RFP/+] mice into microglia-depleted Cx3cr1[CreER/+]R26[DTA/+] mice on days 0, 1, and 2 after TAM administration. Gated on CD11b + Ly6C[−]. Analysis of CNS was performed day 14 after TAM administration. Successful separation of GFP and YFP signals is demonstrated in Supplementary Fig. 3. Values are mean ± s.d of 4 mice/group. The experiment was performed once

expressed in F4/80[hi] macrophages. Notably, several members of the Ms4a cluster located on chromosome 19 (Ms4a7, Ms4a14, Ms4a4a, Ms4a4c, Ms4a6c, Ms4a6b) were also upregulated.

To assess if the unique gene signature in monocyte-derived macrophages resulted in functional changes, we first confirmed increased surface protein expression of Clec12a, CD36, and CXCR4 on F4/80[hi] macrophages compared to F4/80[low] microglia using flow cytometry (Fig. 5e). We next examined phagocytic capacity, since F4/80[hi] macrophages expressed higher levels of scavenger receptors. We observed increased uptake of both E. coli microparticles as well as fluorescently-labelled myelin by F4/80[hi] macrophages compared to in both naive and repopulated F4/80[low] microglia (Fig. 5f). Finally, we assessed their inflammatory cytokine and chemokine

production capacity. Ex vivo isolated F4/80[hi] macrophages secreted similar or lower levels of TNF, CCL3, CCL4 or CCL5 without stimulation or following LPS stimulation compared to F4/80[low] microglia sorted in parallel (Supplementary Fig. 7). We did not detect production of IL-1β, IL-10, IL-12, IFN-γ or CCL2 under these conditions. Our results thus far demonstrated that monocyte-derived macrophages could adopt key components of the microglia transcriptome, but retained large transcriptional and functional differences, even after long-term integration into the CNS.

**Monocyte-derived macrophages adopt microglia DNA methylome.** Epigenetic reprogramming has been demonstrated in

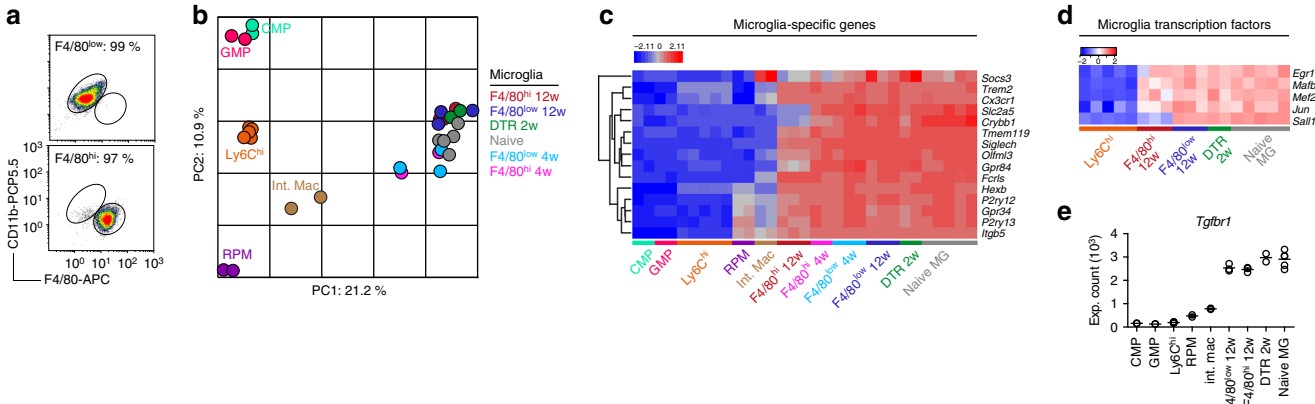

**Fig. 4** Gene expression profile of repopulating microglia/macrophages. **a** FACS purities of sorted F4/80$^{low}$ and F4/80$^{hi}$ populations. **b** PCA of microarray expression profiles. $n$(biological + technical) = 2 GMP, 2 CMP, 3 + 2 Ly6C$^{hi}$, 2 RPM, 2 int. mac, 3 + 2 naive microglia, 2 DTR, 3 F4/80$^{low}$ 4w, 3 F4/80$^{low}$ 12w, 2 F4/80$^{hi}$ 4w, 3 F4/80$^{hi}$ 12w. Naive microglia were sorted from $Cx3cr1^{CreER/+}$ mice that received TAM. DTR microglia were sorted from $Cx3cr1^{CreER/+}R26^{DTR/+}$ mice 14 days after DT administration. F4/80$^{low}$ and F4/80$^{hi}$ subsets in $Cx3cr1^{CreER/+}R26^{DTA/+}$ mice were sorted 4 and 12 weeks after TAM administration. RPM, int. mac, CMP, GMP, and Ly6C$^{hi}$ monocytes were sorted from naive mice. Int. mac. intestinal macrophage. Sorting strategies are detailed in the Methods section. Each sample represents pools of 2–5 mice. **c** Heat map ($z$-scores) of expression profiles of 15 microglia-specific genes. **d** Heat map ($z$-scores) of expression profiles of microglia-specific transcription factors. **e** Microarray expression counts of $Tgfbr1$. Lines represent mean values

macrophages transplanted into a new tissue microenvironment[24]. Furthermore, specific DNA methylation changes occur during hematopoiesis and guide myeloid lineage-specific differentiation[49]. To investigate whether DNA methylation changes occurred in monocyte-derived macrophages we probed the DNA methylome using methylation microarrays. We and others have demonstrated that this technique can be reliably used to probe up to 19,420 CpG sites in the mouse genome[50,51]. We sorted F4/80$^{low}$ and F4/80$^{hi}$ macrophages after 4 and 7 weeks, respectively, and compared their DNA methylomes to naive microglia as well as to BM progenitors and RPMs. Unbiased analysis of all CpG sites organized samples into two major clusters, one containing myeloid progenitors and RPMs, the other containing all CNS-derived samples (Supplementary Fig. 8A). Detailed analysis of the CNS cluster revealed that F4/80$^{low}$ microglia clustered closely with naive microglia, demonstrating that their DNA methylation profile is largely conserved during repopulation (Supplementary Fig. 8A, B). Consistently, we observed only 139 differentially methylated sites between naive microglia and F4/80$^{low}$ microglia at 7 weeks (adj. $p < 0.05$). For F4/80$^{hi}$ macrophages and Ly6C$^{hi}$ monocytes these numbers were 879 and 2081 differentially methylated sites, respectively. Analysis of the most variable sites in the dataset revealed that F4/80$^{hi}$ macrophages adopted a methylation profile distinct from both BM progenitors and their F4/80$^{low}$ counterparts as these three groups occupied the extremes of a multidimensional scaling (MDS) plot visualizing the differences in the dataset (Supplementary Fig. 8B). Contrasting the DNA methylation profiles of these groups using a stringent $p$-value ($p < 0.001$) revealed 1486 sites that were differentially methylated between all groups. Visualization of these CpG sites demonstrated that F4/80$^{hi}$ macrophages adopted an intermediate methylation profile between Ly6C$^{hi}$ monocytes and F4/80$^{low}$ microglia (Supplementary Fig. 8C). These results are consistent with the interpretation that monocyte-derived macrophages partly adopted the DNA methylation profile of microglia following CNS niche colonization.

**Engrafting macrophages adopt a conserved gene signature.** To investigate whether our transcriptomic profiles were biased by our DT-mediated depletion system, we compared our F4/80$^{low}$ and F4/80$^{hi}$ signatures to recently published datasets comparing CNS-engrafting macrophages and microglia. Cronk et al.[52] recently demonstrated that peripheral macrophages engraft the brain as a consequence of impaired microglial self-renewal, using $Cx3cr1^{CreER}+Csf1r^{fl/fl}$ mice. The authors used two additional models to at least partly replace the microglial pool with peripheral macrophages: whole-body irradiation followed by 9 month recovery, or a combination of the CSF1R antagonist PLX5562 and whole-body irradiation. We determined that our F4/80$^{hi}$ gene signature was highly expressed in peripherally-derived macrophages but not locally derived microglia in all three models, and that the opposite was true for our F4/80$^{low}$ gene signature (Fig. 6a).

In another recent study Bennett et al.[53] used intracerebral transplantation (ICT) of microglia or peripheral immune cells into the empty microglial niche in $Csf1r^{-/-}$ mice, demonstrating that only cells of yolk sac origin would adopt bona fide microglial gene expression. We determined that the F4/80$^{hi}$ gene signature was highly expressed in microglia-like cells derived from intracerebrally transplanted BM or blood, but not in intracerebrally transplanted microglia (Fig. 6b). Conversely, the F4/80$^{low}$ gene signature was highly expressed in intracerebrally transplanted microglia, but not in blood or BM-derived microglia-like cells (Fig. 6b).

Taken together these results are consistent with the idea that monocyte-derived macrophages adopt a unique and reliable gene signature as a consequence of encountering the CNS microenvironment. In further support of this conclusion, our F4/80$^{hi}$ and F4/80$^{low}$ gene signatures were conserved in engrafting macrophages and naive microglia reported in two additional studies, one by Bruttger et al.[30] (Fig. 6c) and one by ourselves[31] (Fig. 6d), both using irradiation combined with microglia depletion.

Next we performed gene set enrichment analysis on the transcriptomic profiles of F4/80$^{low}$ and F4/80$^{hi}$ macrophages and naive microglia and visualized their pairwise comparisons using BubbleGum[54]. We initially focused on studies comparing brain-engrafting macrophages with microglia. Overall, the gene sets identified in brain-engrafting macrophages were highly enriched in F4/80$^{hi}$ macrophages, and gene sets identified in CNS resident microglia were enriched in F4/80$^{low}$ and naive microglia (Fig. 6e). This analysis used gene sets from a study comparing the transcriptomic profiles of peripherally-derived and

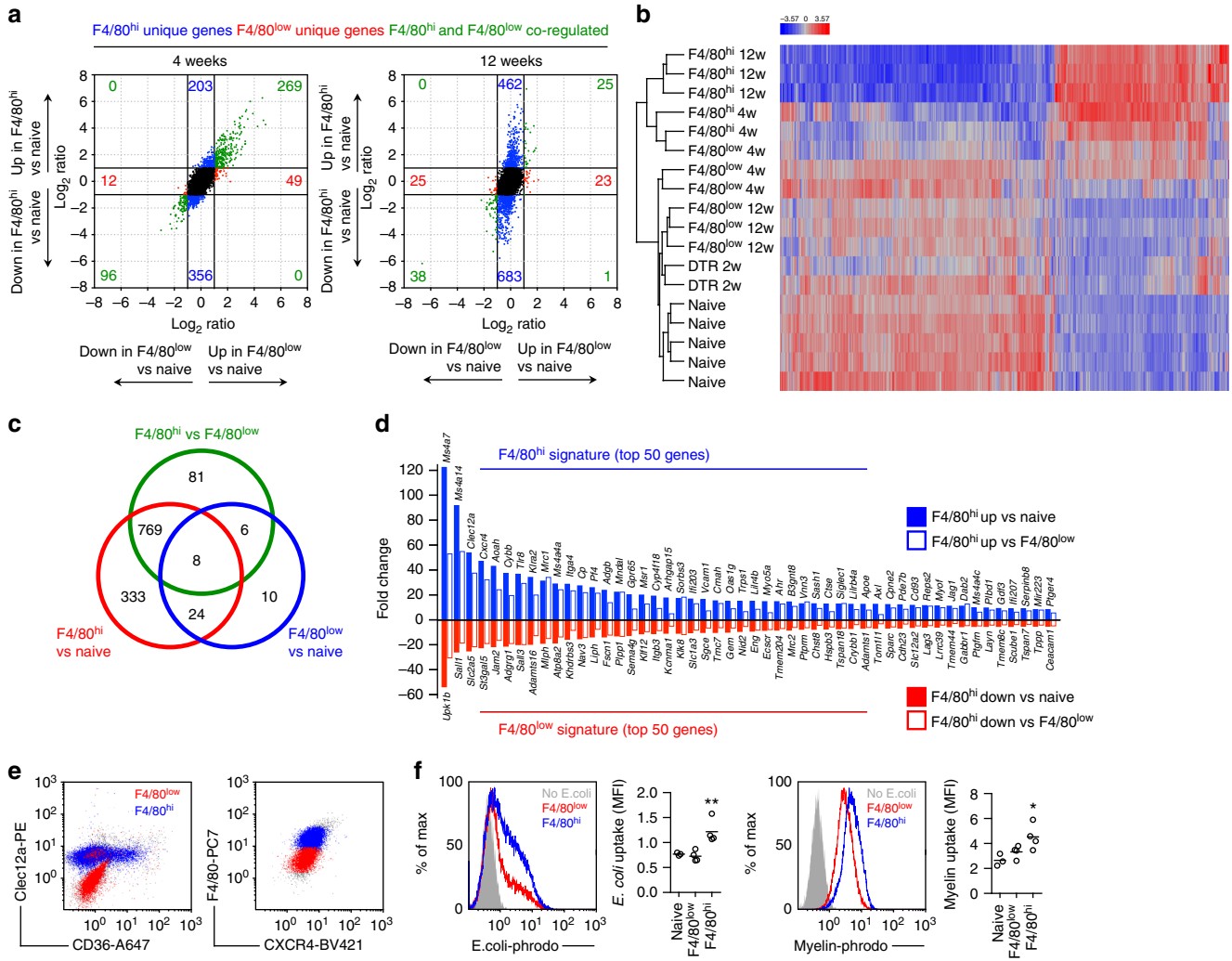

**Fig. 5** Monocyte-derived macrophages are transcriptionally and functionally distinct from resident and proliferating microglia. **a** FC/FC plots comparing differential gene expression in F4/80$^{low}$ and F4/80$^{hi}$ subsets with naive microglia, respectively, after 4 and 12 weeks. **b** Heat map (z-scores) of 1137 genes differentially expressed (>2 fold change both directions, adj. $p < 0.05$) between naive microglia and F4/80$^{hi}$ 12 week subsets. **c** Venn diagram of differential gene expression between naive microglia, F4/80$^{low}$ and F4/80$^{hi}$ subsets at 12 weeks. **d** F4/80$^{hi}$ and F4/80$^{low}$ gene signature; Top 50 up/down-regulated genes in F4/80$^{hi}$ macrophages compared to both naive microglia and F4/80$^{low}$ microglia at 12 weeks. **e** Clec12a, CD36, CXCR4 surface expression in F4/80$^{low}$ and F4/80$^{hi}$ subsets analyzed by flow cytometry. Gated on CX3CR1$^{hi}$CD11b$^+$ cells from *Cx3cr1*$^{CreER/+}$*R26*$^{DTA/+}$ mice > 5 weeks after TAM. **f** Phagocytosis of pHrodo Red *E. coli* microparticles and pHrodo Red-labeled myelin in CD11b-enriched CNS cells, stained with F4/80 to gate on CX3CR1$^+$F4/80$^{low}$ (red) and CX3CR1$^+$F4/80$^{hi}$ (blue) subsets in *Cx3cr1*$^{CreER/+}$*R26*$^{DTA/+}$ mice 5 weeks after TAM. Naive CX3CR1$^+$F4/80$^{low}$ microglia were used as control. Lines represent mean values. *$p < 0.05$, **$p < 0.01$ by Student's paired two-tailed *t*-test (comparing F4/80$^{low}$ and F4/80$^{hi}$)

microglia-derived tumor-associated macrophages in glioma[55], as well as the signature genes in peripherally-derived macrophages (beMφ-50) and microglia (Mg-52) described by Cronk et al.[52] (Fig. 6e). In addition, F4/80$^{hi}$ macrophages displayed enrichment of a common gene signature identified in microglia isolated from mouse models of AD, ALS, and MS, termed neurodegeneration-associated microglia (MGnD)[56] (Fig. 6e). Furthermore, genes differentially expressed in *Sall1* deficient microglia[37] were also highly enriched in F4/80$^{hi}$ macrophages (Fig. 6e), which was of interest since F4/80$^{hi}$ macrophages did not express *Sall1*. Taken together our analysis demonstrates that the F4/80$^{low}$ and F4/80$^{hi}$ gene signatures are conserved across several experimental systems and are unlikely to be biased by our microglia depletion system.

**Type I IFNs regulate macrophage phenotype and colonization.** To further probe potential regulators of peripheral macrophage

engraftment, we performed gene set enrichment analysis on the transcriptional profile of F4/80$^{low}$ and F4/80$^{hi}$ macrophages. We tested for enrichment of the hallmark gene sets, which are 50 well-defined gene sets that convey specific biological states or processes[57] and used BubbleGum to visualize the results. This analysis demonstrated that responses to IFN-γ and to IFN-α were the most significantly enriched biological processes in F4/80$^{hi}$ macrophages (Fig. 7a). This result was further supported by ingenuity pathway analysis performed on the genes differentially expressed in F4/80$^{hi}$ macrophages (>2 fold change both directions compared to F4/80$^{low}$ microglia, adj. $p < 0.05$), which predicted several molecules related to interferon (IFN) signaling (IRF7, IRF3, IFN-α) to most potently affect the F4/80$^{hi}$ gene signature (Fig. 7b). Furthermore, F4/80$^{hi}$ macrophages expressed higher levels of the receptors for type I IFNs, *Ifnar1*, and *Ifnar2*, as well as several IFN stimulated genes, including *Oas1g* and *Oas2* and *Ifi44* (Fig. 7c). These analyses identified type I IFNs as a possible microenvironmental factor shaping F4/80$^{hi}$ macrophages. CNS-

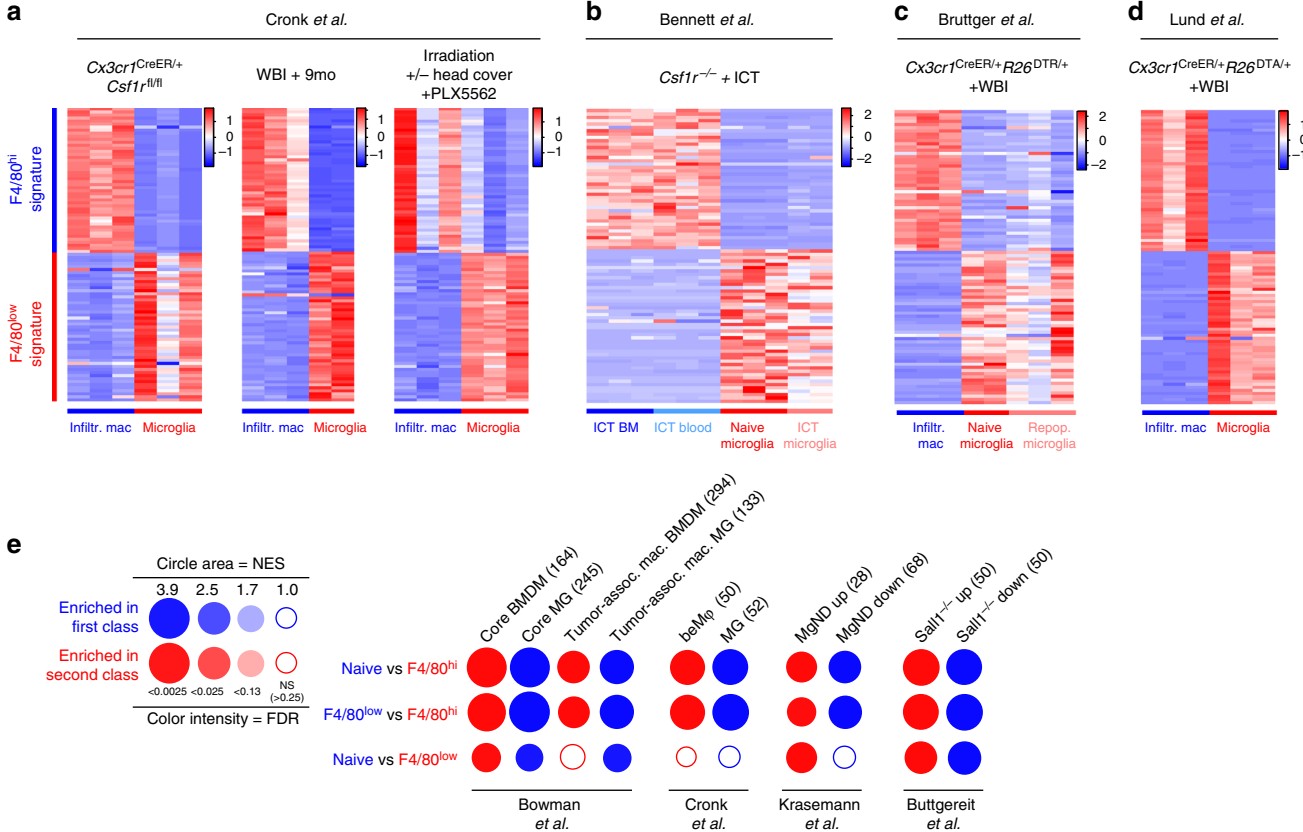

**Fig. 6** Monocyte-derived macrophages adopt a conserved gene signature associated with neuroinflammation and neurodegeneration. **a–d** Expression (z-scores) of the F4/80[low] and F4/80[hi] gene signatures from Fig. 5d in published datasets. These studies were chosen because they compared the transcriptomic profiles of CNS-infiltrating macrophages with resident microglia, with or without microglial depletion. Normalized RNA-seq expression counts were used to plot the data. WBI whole body irradiation. ICT intracerebral transplantation. **e** Gene set enrichment analysis visualized using BubbleGum. Color indicates the cell subset showing enrichment, and the size and color of circles represent enrichment score and significance, respectively. Numbers in parentheses denote the number of genes in the gene set. BMDM bone-marrow derived macrophage MgND neurodegeneration-associated microglia

derived type I IFNs have been demonstrated to regulate infiltrating macrophages during experimental autoimmune encephalomyelitis[58]. We thus asked whether type I IFNs regulate colonization of the microglial niche by monocyte-derived macrophages. To test this experimentally we produced $Ifnar1^{-/-}$: WT → $Cx3cr1^{CreER/+}R26^{DTA/+}$ competitive chimeras, which resulted in approximately 65/35% ($Ifnar1^{-/-}$:WT) chimerism in blood Ly6C[hi] monocytes. We then depleted microglia using TAM administration and analyzed chimerism in the F4/80[hi] compartment after 3 weeks. We observed that $Ifnar1^{-/-}$ cells had a competitive advantage over WT cells in repopulating the F4/80[hi] compartment, resulting in approximately 85/15% ($Ifnar1^{-/-}$:WT) chimerism (Fig. 7d). This indicates that type I IFNs have an inhibitory effect on microglial niche colonization.

## Discussion

Microglia are derived from yolk-sac progenitors that arise early during embryogenesis and that colonize the primitive brain[1]. The formation of the BBB is believed to halt any further colonization of the brain by primitive macrophages during embryogenesis. It is now established that microglia self-maintain throughout adulthood with little or no input from peripheral precursors[1–3], this occurring as a stochastic process of microglial proliferation[59,60]. During certain conditions, however, as demonstrated herein and by others[29,30,52], myeloid cells can colonize the brain and give rise to long-lived microglia-like cells. Understanding how monocytes

take over the microglia niche is not only of biological importance but also of therapeutic interest, since replacement of mutant microglia with wild-type BM-derived cells has been proposed for neurological disorders such as ALS[61,62] and Nasu-Hakola disease[63]. The current study was undertaken to characterize this process and its underlying mechanisms.

In this paper we report that after depletion of microglia using $Cx3cr1^{CreER/+}R26^{DTA/+}$ mice, the microglial niche is simultaneously repopulated by a combination of surviving resident microglia and CNS-infiltrating monocytes. While previous reports have described microglial proliferation or peripheral myeloid cell engraftment to occur as a consequence of microglia depletion[29,30,35,52,64–66], our report is the first that demonstrates that these processes can occur simultaneously and cooperatively to repopulate an empty microglial niche, effectively showing that these processes are not mutually exclusive. Using head-shielded chimeras and adoptive transfer, we could demonstrate that brain irradiation or stem cell release into the circulation were not prerequisites for peripheral macrophage repopulation in our model. Furthermore, monocyte engraftment occurred without BBB breakdown. However, the absence of microglia caused a transient cytokine storm which likely led to the observed vascular activation, coinciding with Ly6C[hi] monocyte entry into the CNS. Future studies should address the importance of this inflammation in attracting monocytes into the CNS.

The fact that proliferating microglia and engrafting macrophages adopted distinct surface marker expressions (F4/80[low] and

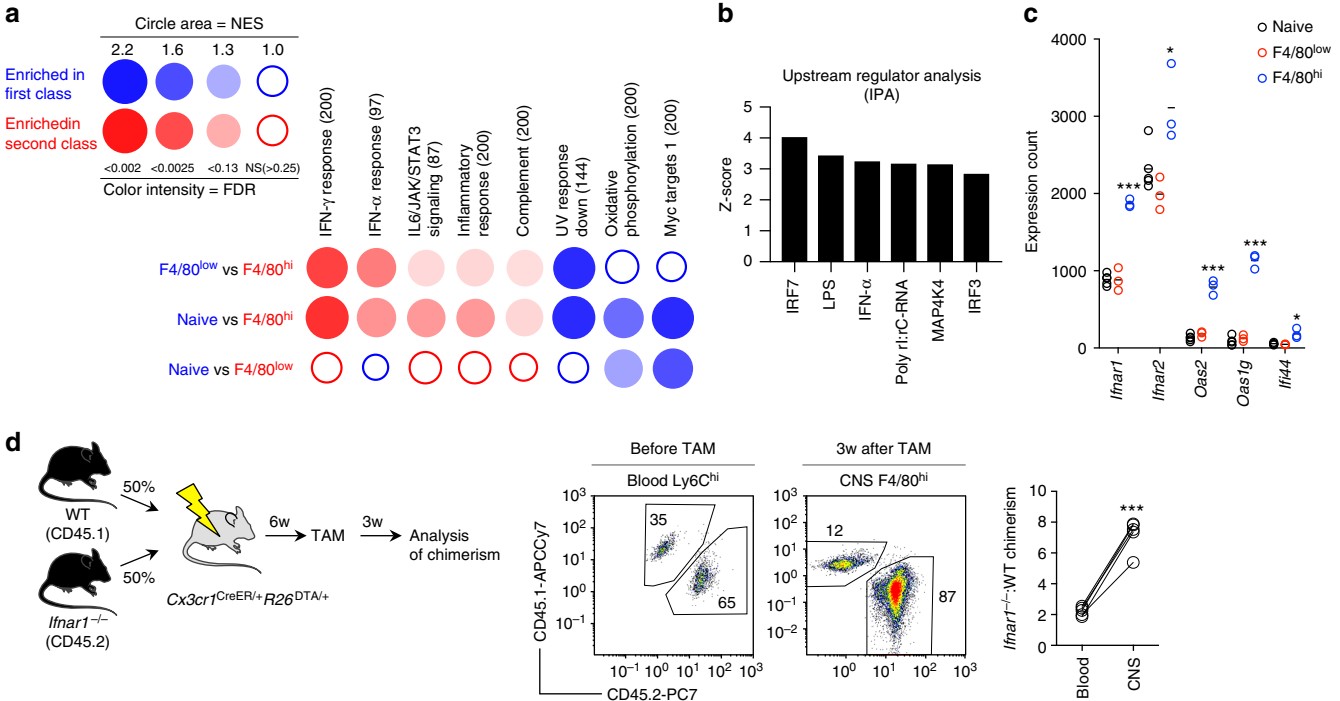

**Fig. 7** Type I IFNs regulate niche colonization. **a** Analysis of enrichment of hallmark gene sets visualized using BubbleGum. Color indicates the cell subset showing enrichment, and the size and color of circles represent enrichment score and significance, respectively. **b** Top activated upstream regulators using ingenuity pathway analysis. **c** Microarray expression counts of type I IFN receptors and response genes. Lines represent mean values. *$p < 0.05$, ***$p < 0.001$ by Student's unpaired two-tailed $t$-test comparing F4/80[low] and F4/80[hi]. **d** Analysis of chimerism in WT:Ifnar1[−/−] competitive chimeras before TAM administration (blood) or 3 weeks after TAM (CNS). Blood is gated on CD11b[+]SSC[low]Ly6C[hi] monocytes and CNS on CD11b[+]F4/80[hi]. $n = 4$ mice. The experiment was performed once. ***$p < 0.001$ by Student's paired two-tailed $t$-test

F4/80[hi], respectively) could be utilized to sort these populations and perform transcriptional, epigenetic, and functional profiling. One obvious benefit of our experimental paradigm is that we could thus control for the effect of the CNS environment during depletion/repopulation, since F4/80[low] and F4/80[hi] cells were sorted from the same brains. Consequently, our first observation was that the global gene expression profile of CNS-engrafting macrophages was more similar to microglia than to other tissue-resident macrophages or their Ly6C[hi] monocyte precursors. Consistent with this we observed upregulation of microglia-specific genes including *P2ry12*, *Fcrls*, *Siglech*, and *Tmem119* in monocyte-derived macrophages, as well as microglia-expressed transcription factors *Mafb* and *Mef2a*. In addition, we observed a similar trend for the DNA methylation profile. In concordance with a recent proposal,[28] we interpret this as imprinting of monocyte-derived macrophages by the microglial niche. Importantly, however, *Sall1*, a transcription factor regulating microglia identity[37] was not expressed in monocyte-derived macrophages, whose transcriptional profile displayed enrichment of genes expressed in *Sall1*[−/−] microglia. Enforced expression of *Sall1* in CNS-engrafting macrophages would be an important experiment to address the requirement for *Sall1* expression in coordinating the bona fide microglia gene expression program.

Our F4/80[low] and F4/80[hi] gene signatures consistently overlapped with the published transcriptional profiles of microglia and engrafting macrophages across several models of microglia depletion, ruling out the possibility that they were biased by the nature of our microglia depletion model. Rather, the conserved gene signature observed in F4/80[low] and F4/80[hi] subsets instead indicates that these gene sets represent fixed ontogeny-dependent expression programs. We consistently observed a high similarity between our F4/80[hi] gene signature and that described in

microglia-like cells derived from intracerebral transplantation of cells of hematopoietic stem cell origin into an empty microglial niche[53], including high expression of *Ms4a7*, *Clec12a*, and *Apoe*. This was in contrast to cells of yolk-sac origin[53], which displayed higher expression of microglia-specific genes, including *Sall1*, *Sall3*, and *Slc2a5*.

Why do monocyte-derived macrophages not fully adopt the microglia gene expression program despite being situated in the same organ space? One possibility is that microglial education occurring during development[46,67] induces a transcriptional and epigenetic program not obtainable by monocyte-derived macrophages that enter the adult brain. Alternatively, monocytes may contain unique epigenetic programming from the BM. In concordance with this notion we observed that the DNA methylation pattern of monocyte-derived macrophages was frozen between BM and microglia states. Interestingly, Lavin et al. have demonstrated considerable variation in the ability of BM-derived transplants to recover tissue macrophage-specific chromatin states. Depending on the tissue, transplant-derived macrophages isolated from the liver, spleen, lung or peritoneum adopted 47–92% of tissue-specific enhancer regions[24]. Such variability could potentially explain why monocytes do not fully adopt microglia gene expression, which contrasts to monocytes repopulating Kupffer cell[68] or alveolar macrophage[26] niches following experimental depletion.

Finally, to identify potential regulators of monocyte engraftment and repopulation we performed gene set enrichment and ingenuity pathway analyses, which identified a robust IFN signature in monocyte-derived macrophages. Importantly, type I IFNs regulate the phenotype of both CNS resident and infiltrating myeloid cells. Exaggerated type I IFN signaling in microglia leads to their activation and engulfment of synapses, manifesting as

neurological symptoms in lupus-prone mice[69]. However, local production of type I IFNs in the CNS suppress autoimmune neuroinflammation by modulating CNS engrafting macrophages[58]. We used a competitive chimera approach to address the requirement for IFNAR signaling in monocyte-derived macrophages in repopulating the microglial niche, and demonstrated that type I IFNs impair this process. Interestingly, this contrasts with our recent observation that TGF-β signaling promotes microglia niche colonization[31]. Future studies should aim to identify the crucial microenvironmental factors shaping CNS engrafting monocytes and how they may be manipulated in order for monocytes to completely adopt microglia identity.

## Methods

**Mice**. All mice were bred and maintained under specific pathogen-free conditions at Karolinska Institutet, in accordance with national animal care guidelines. All animal experiments were approved by the appropriate ethical review board (Stockholms djurförsöksetiska nämnd). $Cx3cr1^{CreER-EYFP}$, $R26^{DTR}$, $R26^{DTA}$, CD45.1 were originally obtained from The Jackson Laboratory. $Cx3cr1^{GFP/+}Ccr2^{RFP/+}$, and $Ccr2^{-/-}$ $^{(RFP/RFP)}$ mice were a gift from Klas Blomgren at Karolinska Institutet. $Ifnar1^{-/-}$ were originally obtained from Ulrich Kalinke. Experiments were started when mice were 6–12 weeks.

**In vivo treatments**.
Tamoxifen administration: Tamoxifen (TAM; Sigma) was suspended in corn oil at 75 °C and 5 mg (200 μl) was unless otherwise specified administered subcutaneously (s.c.) on three consecutive days.

Diphtheria toxin administration: Microglia were depleted in $Cx3cr1^{CreER/+}R26^{DTR/+}$ by two subcutaneous doses of 5 mg TAM separated by 48 h followed 3 weeks later by three daily doses of 25 ng/g diphtheria toxin, given intraperitoneally.

EdU administration: Mice were fed for 2 weeks with EdU in drinking water (0.2 mg/ml + 1% sucrose) from days 0–14 or 14–28 after TAM injections. The solution was protected from light at all times and changed once per week. EdU was detected by flow cytometry according to manufacturer's instructions (Invitrogen).

Evans Blue administration: To assess BBB integrity mice were injected i.p with 200 μl 2% Evan's Blue solution (Sigma). Twenty-four hours later mice were perfused and brains collected, weighed and homogenized in 1 ml 3.05 M trichloroacetic acid. The homogenate was centrifuged for 10 min at $10,000 \times g$ and the supernatant mixed 1:3 with 95% ethanol. Samples were excited at 625 nm and emission read at 680 nm using a Glomax fluorescence plate reader.

**Generation of BM chimeras**. Mice were irradiated with 9.5 Gray using an X-RAD 320 irradiation source (0.95Gray/minute) with a $20 \times 20$ cm irradiation field. Head protection was accomplished by retaining mice under isofluorane anaesthesia and placing the head (from the neck up) outside the field of irradiation. Mice were monitored throughout the irradiation period to make sure the heads stayed outside the field (the irradiation source is equipped with a lamp to visualize the irradiation field). Mice were reconstituted on the same day with $2–5 \times 10^6$ BM cells by tail vein injection. Mice were considered reconstituted and used for experiments 6–8 weeks later. Head-protected mice generally resulted in 50–80% chimerism.

**Chemotherapy**. Myeloablation using chemotherapy was performed by administering 20 mg/kg Busulfan (Sigma) by i.p. injection on three consecutive days. Busulfan was solubilized in DMSO and diluted to 5 mg/ml in PBS. On the fourth day $12–14 \times 10^6$ BM cells were given by tail vein injection. This procedure resulted in 80–98% chimerism.

**Adoptive transfer**. $Ly6C^{hi}$ monocytes were isolated with ~90% purity using negative selection MACS beads (monocyte isolation kit, Miltenyi) from the femurs and tibias of $Cx3cr1^{GFP/+}Ccr2^{RFP/+}$ mice. $1.7–3.0 \times 10^6$ cells/day were transferred by tail vein injection into $Cx3cr1^{CreER/+}R26^{DTA/+}$ mice on day 0, 1, and 2 after TAM. Mice were sacrificed on day 14 for analysis of reconstitution.

**Preparation of single cell suspensions**. Mice were sacrificed by injecting 100 μl pentobarbital (i.p.). When applicable, blood was collected from the right ventricle prior to perfusion. Mice were perfused with ice- cold PBS and organs dissected. CNS cells were prepared by enzymatic digestion using Collagenase D (1 mg/ml, Roche) and DNAse I (0.2 mg/ml, Roche) or Neural Tissue Dissociation Kit T (Miltenyi Biotec). Myelin was removed using 38% Percoll.
Samples of 100–200 μl blood were collected into tubes containing EDTA, lysed in ACK buffer and centrifuged. The pellet was resuspended in PBS and used for staining. Spleen cell suspensions were prepared by mechanical dissociation in PBS using 40 μm strainers. BM cells were prepared by flushing femurs with PBS. Spleen

and BM preparations were treated with ACK buffer to lyse RBCs. Cell suspensions from the lamina propria of the small intestine were prepared by mechanical dissociation followed by enzymatic digestion using DNAse and Liberase. Cells were then purified using a 40/60% Percoll gradient and isolated at the interphase. Cells were counted using a Scepter counter (Millipore) or by flow cytometry using CountBright absolute counting beads (Thermo Fisher).

**Flow cytometry**. Single cell suspensions were plated in 96-well V-bottom plates and stained at 4 °C. Dead cells were removed using LIVE/DEAD Fixable Dead Cell Stain Kit (Invitrogen). The following antibodies were used: CD3 (17A2, Biolegend), CD11b (M1/70, Biolegend), CD11c (N418, Biolegend), CD16/32 (93, Biolegend), CD34 (HM34, Ebioscience), CD36 (MF3, Bio-Rad), CD45 (30F11, Biolegend), CD45.1 (A20, Biolegend), CD45.2 (104, Biolegend), CD115 (AFS98, Biolegend), CD206 (MR5D3, BD), B220 (RA3-6B2, Biolegend), c-kit (ACK2, Biolegend), Clec12a (5D3CLEC12A, Biolegend), CXCR4, (2B11, BD), F4/80 (BM8, Biolegend), Ly-6C (HK1.4, Biolegend), Ly-6G (1A8, BD Biosciences), MHCII (M5/114.15.2, Biolegend), NK1.1 (PK136, BD Biosciences). Sca-1 (D7, Biolegend), Siglec H (551, Biolegend), TER119, (TER-119, Biolegend). Cells were acquired using a Gallios flow cytometer (Beckman Coulter) and analyzed using Kaluza software (Beckman Coulter). GFP and YFP signals in chemotherapy and adoptive transfer experiments were separated by exciting GFP using the 405-nm violet laser using the 550/40-nm emission filter.

**Cell sorting**. Cells were sorted using a BD influx cell sorter using the following sorting strategies. $F4/80^{low}$ microglia ($CD11b^+CX3CR1^{hi}F4/80^{low}$), $F4/80^{hi}$ macrophages ($CD11b^+CX3CR1^{hi}F4/80^{hi}$), Naive microglia and $Cx3cr1^{CreER/+}R26^{DTR/+}$ (DTR) microglia ($CD11b^+CX3CR1^{hi}$) with >97% purity. CMP ($Lin^-ckit^+sca1^-CD34^+CD16/32^{int}$), GMP ($Lin^-ckit^+sca1^-CD34^+CD16/32^{hi}$), cMoP ($Lin^-ckit^+CD115^+Ly6C^+CD11b^-$), $Ly6C^{hi}$ monocytes were sorted from BM using the monocyte isolation kit (BM, Miltenyi Biotec) with ~95% purity. Spleen RPMs were sorted based on their superparamagnetic properties using MACS columns with ~95% purity or further FACS sorted ($Dead^-CD45^+F4/80^{hi}$). Intestinal macrophages were pre-sorted using CD11b-beads (Miltenyi) and then as $CX3CR1^{hi}CD64^+MHCII^+$.

**Microarray analysis**. For RNA preparation cells were sorted into a solution of RNA later (Thermo Fisher Scientific). RNA was prepared using RNeasy Micro Kit (Qiagen). RNA quality and integrity was assessed using a Bioanalyzer 2100 (Agilent). All samples included had high quality (RIN = 7–10). cDNA preparation, hybridization and scanning were performed at the Array and Analysis Facility, Science for Life Laboratory at Uppsala Biomedical Center (BMC). 0.5–10 ng of total RNA from each sample was used to generate amplified and biotinylated sense-strand cDNA from the entire expressed genome according to the GeneChip WT Pico Reagent Kit User Manual (Affymetrix). GeneChip Mouse Gene 2.1 ST Arrays were hybridized for 16 h in a 45 °C incubator, washed and stained and finally scanned at the GeneTitan Multi-Channel Instrument, according to the GeneTitan Instrument User Guide for Expression Arrays Plates (Affymetrix).

**Bioinformatic analyses**. PCA and heatmaps were produced using Partek Genomics Suite (Partek).
Gene Set Enrichment analysis was performed and visualized using BubbleGum using standard settings. Ingenuity pathway analysis (IPA, Qiagen) was used to analyze upstream regulators.

**DNA methylation analysis**. DNA was prepared using QIAamp DNA Micro Kit (Qiagen). DNA methylation analysis was performed using Infinium MethylationEPIC BeadChip (Illumina) with ≥250 ng input DNA at the Bioinformatics and Expression Analysis (BEA) core facility, Karolinska Institutet. EPIC probes (50-mer oligonucleotides) were mapped to GRCm38/mm10 using Bismark (version 0.14.5) with default settings. A total of 19,420 probes were identified as unique hits (conserved between human and mouse) and used for downstream analysis. Data was analyzed using the ChAMP package, probes were filtered for mapping probes with 17,633 probes passing detection cutoff, and subsequently SWAN normalized. Differential probes were identified using Limma ($p < 0.01$) and clustering was preformed using PCA analysis.

**qPCR**. Hemibrains or coronal brain sections were homogenized in RLT-buffer (Qiagen) containing beta-mercaptoethanol (14.3 M, 10 μl/ml, Sigma) and RNA prepared using the RNeasy Mini Kit (Qiagen). cDNA was synthesized using the iScript cDNA Synthesis Kit (Bio-Rad). qPCR was performed with SYBR green reaction (Bio-Rad). All expression levels are reported relative to *Hprt* or *Hprt* and *Gapdh*. Primer sequences can be found in Supplementary Table 1.

**Immunohistochemistry**. PBS-perfused brains were immersion-fixed in 4% PFA for 24 h, then sucrose protected (20%) for at least 24 h and then embedded in OCT cryomount (Histolab) and frozen in isopentane. Sections were stained using the following antibodies: Iba-1 (Wako), F4/80-biotin (Cl:A3-1, AbD Serotec), GFP (Abcam, ab13970), GFAP (Abcam, ab7260), Ki67 (Abcam, ab15580), P2ry12

(generated by Dr. O. Butovsky), ICAM-1 (Abcam, ab119871). P2ry12-staining and CX3CR1[+]/Ki67[+] cell counting was performed Pannoramic viewer and Histoquant software (3D histech).

### Ex vivo experiments.

Phagocytosis: Phagocytosis was assessed in F4/80[low] and F4/80[hi] cells by pre-sorting CNS cells using CD11b-microbeads (Miltenyi) and staining with F4/80 and CD11b antibodies. Cells were then incubated 1 h with pHrodo Red E. coli microbeads (Thermo Fisher Scientific) or pHrodo Red-labeled myelin. Myelin was isolated from mouse brains/spinal cords using a sucrose gradient and labeled with pHrodo Red according to manufacturer's instructions (Thermo Fisher Scientific).

Cytokine secretion: Sorted microglia were cultured overnight in DMEM containing 10% FCS ± 100 ng/ml LPS (E. coli O111:B4, Sigma). Culture supernatants were analyzed for levels of TNF, IFN-γ, IL-1β, IL-10, IL-12p70, CCL2, CCL3, CCL4, and CCL5 in two multiplex-panels using cytometric bead array (CBA) according to manufacturer's instructions (BD biosciences).

### Data availability

The data that support the findings of this study are available from the corresponding author upon reasonable request. Microarray and DNA methylation array raw data have been deposited in the Gene Expression Omnibus data repository under accession number GSE121409 and GSE121483, respectively. A reporting summary for this article is available as a Supplementary Information file. The source data underlying Figs. 5d, 6a–e, 7a, b are provided as source data files.

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

## Acknowledgements

We thank Dr. Annika van Vollenhoven for FACS sorting. We thank Elisabeth Qvist for animal care-taking and Michelle Gustafsson for performing tail vein injections. We would like to acknowledge the Array and Analysis Facility, Science for Life Laboratory at Uppsala Biomedical Center (BMC) for performing microarray analysis. DNA methylation analysis was performed at the Bioinformatics and Expression Analysis core facility, Karolinska Institutet, Huddinge. Figures containing mice graphics (Figs. 2, 3 and 7) were produced using Servier Medical Art (http://smart.servier.com/). This work was supported by grants from the Swedish Alzheimer Foundation (Alzheimerfonden, R.A.H.), the Swedish Research Council (R.A.H. and M.J.), the Swedish Brain Foundation (Hjärn-fonden, R.A.H.), Alltid Litt Sterkere (R.A.H.) and a fellowship from the Margaretha af Ugglas Foundation (L.K.).

## Author contributions

H.L. conceived the study with R.A.H., X.M.Z., and R.P. H.L., R.P., R.A.H., X.M.Z., O.B. designed experiments. H.L., M.P., R.P., D.G., and X-M.Z. performed most experiments and analyzed the data. Additional experiments/data analysis or design were performed by: J.H. (flow cytometry) L.K. (DNA methylation) E.E., M.N., M.J. (bioinformatic analyses) A.E., E.N., A.K.Ö. (Ifnar1 experiments). H.L. wrote the paper which all co-authors edited.

## Additional information

**Competing interests:** The authors declare no competing interests.

