## [Peer Review File · Nature Communications]

Reviewers' comments:

Reviewer #1 (microglia, brain-immune)(Remarks to the Author):

In the research paper by Lund et al., the authors used CX3CR1CreER/+R26DTA/+ mice to deplete microglia in the brain by tamoxifen-inducible intracellular diphtheria toxin A (DTA) expression and show that repopulating cells represent a mixed population of resident microglia and blood-borne macrophages. Once recruited into the brain, Ly6Chi monocytes acquire a microglia-like gene expression profile, whilst they keep a partially distinct gene signature compared to microglia. This is associated with partially different DNA methylation signatures, phagocytic capacity and cytokine production.

Recent studies, including a paper by the authors have used diphtheria toxin-based approaches to deplete microglia (e.g. Bruttger et al., *Immunity* 2015; Lund et al., *Nat Immunol* 2018). It has also been demonstrated that peripheral-derived cells can engraft the brain to replace microglia (e.g. Bruttger et al., *Immunity* 2015; Cronk et al., *J Exp Med*, 2018, Lund et al., *Nat Immunol* 2018), although these reports have come to partially different conclusions regarding the proportions of repopulating resident versus exogenous cells, the need for irradiation for the entry of exogenous cells and the mechanisms involved. It has also been demonstrated that repopulating macrophages / monocytes maintain a unique functional and transcriptional identity as compared to microglia (Cronk et al., *J Exp Med*, 2018).

The interesting novel finding presented in the paper is that Ly6Chi monocytes, which – together with resident, proliferating microglia - contribute to the repopulation process without the involvement of hematopoietic progenitors, upregulate genes that are normally attributable to microglia and adopt microglia-like DNA methylation signatures. These data suggest that the brain microenvironment would instruct recruited cells to become functionally similar to microglia, although these cells would keep their initial transcriptomic phenotype and distinct phagocytic, inflammatory properties. These results are valuable for the field since they suggest that the replacement of dysfunctional microglia in a number of human diseases could be therapeutically feasible.

One potentially important weakness of the DTA model used in this context is that the mechanisms of microglial cell death have not been assessed / discussed and based on the data presented it is likely that the depletion process triggers inflammation in the brain (more than seen in the case of microglia depletion by CSF1R antagonists). The effect of these inflammatory changes on the transcriptomic signatures of the repopulating cells has not been appropriately characterized. The authors show that despite BBB injury is not detectable, there is a considerable increase in cytokines / chemokines, whilst no information is available on how the vasculature and other cells in the neurovascular unit have been influenced by microglia depletion. Even 28 days after depletion, the distribution of repopulating cells look uneven, and cells are morphologically activated (Fig.1D). These data suggest that microglia depletion by DTA in itself could shape the transcriptomic profile of both resident, repopulating microglia and determine the identity and transcriptomic features of cells recruited from the periphery. Interestingly, in the case of microglia depletion by CSF1R blockade, the contribution of blood-borne monocytes / macrophages to repopulation is minor (e.g. Huang et al., *Nat Neurosci*, 2018). It has also been shown that peripheral-derived macrophages can engraft the brain independent of irradiation in Cx3cr1CreER/+::Csf1rFlox/Flox mice, whilst whole-body irradiation was required for considerable exogenous cell infiltration after microglia have been eliminated by PLX5622 (Cronk et al., *J Exp Med*, 2018). Therefore, from the present paper it remains unclear whether transcriptomic signatures of recruited Ly6Chi monocytes that turn into F4/80hi brain macrophages would acquire identical transcriptomic and methylation signatures if the brain microenvironment was less proinflammatory after microglia depletion. It is suggested that the authors investigate whether repopulating

macrophages by using another model of microglia depletion (e.g. repopulation after PLX5622-induced depletion combined with whole-body irradiation or adoptive transfer of Ly6Chi monocytes). This way it is possible to assess to what extent infiltrating cells keep their macrophage identity or become similar to microglia with special attention to transcriptomic changes in the otherwise „microglia-specific“ markers (e.g. P2Y12, Siglec H, Tmem119, Sall1, etc).

Further points:

- CD11b levels appear to be upregulated and remain high in resident microglia even several weeks after repopulation, in contrast to lower levels seen in F4/80hi cells. How does this compare to CD11b levels seen in brain-infiltrating cells before they enter the brain?
- Analysis of Ly6Clow monocytes in the blood during depletion (Suppl. Fig. 2F) suggests a reduction of these cells in response to tamoxifen administration. Please explain. Was the actual „0 day“ sample taken immediately after the 3rd tamoxifen treatment?
- At d28 there seems to be an increase in CX3CR1- / F4/80- cells in the brain (Fig 1D). Have the authors observed a recruitment of granulocytes, T cells or B cells into the brain after microglia depletion?
- Monocyte recruitment is expected to be associated with the upregulation of vascular adhesion molecules – it would be useful to show whether microglia depletion contributed to vascular activation in these studies or the route of cell infiltration into the brain parenchyma was primarily via the meninges / choroid plexus.
- Whilst resident microglia appear to return to a more homeostatic state 12 weeks after depletion (4 weeks after tamoxifen administration 361 differentially expressed genes compared to naive microglia, and 56 genes after 12 weeks), recruited blood-borne macrophages appear to increase their differentially expressed genes over time (886 and 1173 differentially expressed genes, 4 weeks and 12 weeks after depletion, respectively). This difference is also visible in the heat maps presented in Fig.5D. Do the authors have an explanation to this? Is it possible that unlike in the case of long-lived microglia, recruited macrophages die over time and are replaced by additional waves of blood-borne monocytes several weeks after depletion?

Reviewer #2 (immune-neuro crosstalk, NK)(Remarks to the Author):

In this work the authors are demonstrating that microglia can be repopulated by peripheral monocytes if there is a niche/room provided through inability of microglia to proliferate and fill the niche. The manuscript is very well done and is addressing an important topic. The results are somewhat novel compared and controversial to decades of previous studies suggesting monocytes cannot repopulate microglia unless the brain is irradiated. However, in the last few months two papers have appeared which are showing very similar findings, which to some extent erode the novelty of this finding. The previous works (Cronk et al., JEM 2018 and Bennett et al Neuron 2018) are not cited or discussed here. Especially the JEM paper is virtually showing very similar things to these findings. It would be interesting if the authors have more closely compared their findings to those made previously. The authors are using fewer models and less advanced bioinformatics than the other two studies, but they get into more nuance by showing that the monocyte-derived macrophages take on some of the properties of microglia, yet ultimately maintain a unique identity. Overall, this is an interesting story and upon minor revision addressing the comments above, it would

make a strong contribution to the current literature on microglia.

Reviewer #3 (epigenetic)(Remarks to the Author):

I have read the paper by Lund et al. with interest since I am somewhat familiar with the macrophage/niche debate even though this is not a main theme in my own research. The paper is well written and, from my point of view, convincingly argued. At the Editor's request, I paid particular attention to the epigenetic/DNA methylation data, and I now understand why some perplexity may have arisen. In essence, the authors used the brand new Illumina EPIC platform that interrogates almost 900,000 HUMAN CpGs to survey the mouse macrophage/glia methylome. As I understand it, the use of a platform designed for a species that diverged from human 80 millions years ago is the issue.

The authors have thought about this issue, and in fact they published a 2017 paper which directly addresses the possibility of using the EPIC platform for mouse methylome studies. The conclusions of this paper (Needhamsen et al, ref. 59) are self-explanatory:

"We identified 19,420 EPIC probes (referred as mEPIC probes), which align with a unique best alignment score to the bisulfite converted reference mouse genome mm10. Further annotation revealed that 85% of mEPIC probes overlapped with mm10.refSeq genes at different genomic features." And later:

"Overlap analysis revealed that approximately 84% (16,352 out of 19,420) of mEPIC probes overlapped with annotated mm10.refSeq genes, but that each gene was targeted by only a few probes, therefore limiting the use of EPIC for certain applications such as detection of differentially methylated regions (DMRs). Nevertheless, the utility of EPIC array for DNA methylation analysis using mouse samples remains suitable for broader applications such as cluster analysis."

In other words, this approach can detect some mouse CpGs throughout the genome with a reasonably even distribution, but can interrogate only a minority of CpG sites and only a few CpGs per gene – basically putting us in the uncomfortable position in which the field was a few years ago when Illumina only had the 27K chip, which became rapidly obsolete.

In my view, these data add little to the paper and its message and in fact to some extent they weaken it technically. At best, if my CNS expert co-reviewers consider this work worth publishing, I would advise relegating the DNA methylation data to the Supplemental Material, toning down the relevant section in the text and making it extremely clear that this analysis is purely exploratory.

Response to Reviewers' comments:

Reviewer #1 (microglia, brain-immune):

1. One potentially important weakness of the DTA model used in this context is that the mechanisms of microglial cell death have not been assessed / discussed and based on the data presented it is likely that the depletion process triggers inflammation in the brain (more than seen in the case of microglia depletion by CSF1R antagonists). The effect of these inflammatory changes on the transcriptomic signatures of the repopulating cells has not been appropriately characterized. The authors show that despite BBB injury is not detectable, there is a considerable increase in cytokines / chemokines, whilst no information is available on how the vasculature and other cells in the neurovascular unit have been influenced by microglia depletion. Even 28 days after depletion, the distribution of repopulating cells look uneven, and cells are morphologically activated (Fig.1D). These data suggest that microglia depletion by DTA in itself could shape the transcriptomic profile of both resident, repopulating microglia and determine the identity and transcriptomic features of cells recruited from the periphery. Interestingly, in the case of microglia depletion by CSF1R blockade, the contribution of blood-borne monocytes / macrophages to repopulation is minor (e.g. Huang et al., Nat Neurosci, 2018). It has also been shown that peripheral-derived macrophages can engraft the brain independent of irradiation in Cx3cr1CreER/+::Csf1rFlox/Flox mice, whilst whole-body irradiation was required for considerable exogenous cell infiltration after microglia have been eliminated by PLX5622 (Cronk et al., J Exp Med, 2018). Therefore, from the present paper it remains unclear whether transcriptomic signatures of recruited Ly6Chi monocytes that turn into F4/80hi brain macrophages would acquire identical transcriptomic and methylation signatures if the brain microenvironment was less proinflammatory after microglia depletion. It is suggested that the authors investigate whether repopulating macrophages by using another model of microglia depletion (e.g. repopulation after PLX5622-induced depletion combined with whole-body irradiation or adoptive transfer of Ly6Chi monocytes). This way it is possible to assess to what extent infiltrating cells keep their macrophage identity or become similar to microglia with special attention to transcriptomic changes in the otherwise “microglia-specific” markers (e.g. P2Y12, Siglec H, Tmem119, Sall1, etc).

We understand that this reviewer's major concern is that the microglial cell death triggers an inflammatory process that drives the transcriptomic signature observed in repopulating F4/80^{low} and F4/80^{hi} cells. While there is definitely evidence of a 'cytokine storm' directly following depletion, this has largely abated by day 14 (Fig. R1). We have performed additional qPCR analyses for several inflammatory cytokines and chemokines and this new data is now included in Supp. Fig. 2C in the revised manuscript.

Figure R1: mRNA levels of cytokines and chemokines 7, 14 and 28 days after TAM administration.

In addition, per the reviewer's suggestion, we have assessed vascular activation through immunohistochemical staining for ICAM-1 and the results are summarized under the specific question below (Fig. R7). We interpret these new data to mean that there is indeed an inflammatory reaction directly following depletion, which likely attracts monocytes to the brain. However, the inflammation does not persist long-term, and we therefore do not expect it to determine the transcriptional state of F4/80^{low} and F4/80^{hi} cells. We have also performed several analyses to control for the effect of this inflammation, and the results are summarized below.

To control for the inflammatory environment, the reviewer suggests depleting microglia using PLX5622 combined with whole-body irradiation (WBI) and to compare the gene signature of infiltrating cells with repopulating microglia. However, this exact experiment is already in the public domain (Cronk *et al* 2018, JEM), an article cited by the reviewer. This therefore allows us to compare the gene signature of infiltrating macrophages and locally proliferating microglia in our model with those of the PLX5622 model. In addition, we extended this comparative data set analysis to include several other recently published models of microglia depletion/peripheral repopulation, including Cx3cr1^{CreER}Csf1^{fl/fl} (Cronk *et al*), whole body irradiation (Cronk *et al*), Cx3cr1^{CreER}R26^{DTR} with whole body irradiation (Bruttger *et al* 2015, Immunity), and intracerebral transplantation (ICT) of macrophages or microglia into microglia-deficient Csf1^{-/-} mice (Bennett *et al* 2018, Neuron). Through these comparative analyses we determined that our F4/80^{hi} signature genes were consistently upregulated in the infiltrating macrophages in all of these models, and that our F4/80^{low} signature genes were consistently upregulated in naive microglia or repopulating microglia (Fig. R2). We consider that this analysis is compelling and does not warrant additional experimentation at this time. This comparative transcriptomic data has been included as a new Fig. 6 in the revised manuscript.

Figure R2: (A) Top 50 up/down regulated genes in infiltrating F4/80^{hi} macrophages vs resident proliferating F4/80^{low} microglia. **(B)** Expression (z-scores) of the F4/80^{low} and F4/80^{hi} gene signatures from published datasets. **(C)** Enrichment of published gene sets in the F4/80^{low} and F4/80^{hi} transcriptomic signatures using BubbleGum. Color indicates the cell subset showing enrichment, and the size and color of circles represent enrichment score and significance, respectively.

In addition, we consider that one specific advantage of our mixed repopulation system is that gene expression in infiltrating F4/80^{hi} cells can be compared to repopulating F4/80^{low} microglia as a control, directly addressing the concern that neuroinflammation may alter macrophage gene expression. It is clear from published datasets that inflammation dramatically changes the transcriptional state of native microglia (Bennett *et al* 2016, PNAS). If the depletion gave rise to inflammation that causes long-lasting transcriptional changes, one would therefore expect to see them in both F4/80^{low} and F4/80^{hi} cells at 12 weeks. We find that despite exposure to the same brain signals, F4/80^{hi} cells have a gene expression profile that is distinct from neighboring F4/80^{low} microglia. We have highlighted this in greater detail by including a Venn diagram in Fig. 5C in the revised manuscript, which illustrates a comparison

of naive microglia, proliferating F4/80^{low} microglia and infiltrating F4/80^{hi} macrophages at 12 weeks (Fig. R3).

Figure R3: Comparison of repopulating F4/80^{low} and F4/80^{hi} macrophages at 12 weeks with naive microglia.

Finally, from the gene set enrichment analysis comparing F4/80^{low} and F4/80^{hi} macrophages, one factor that seems to shape the transcriptomic signature of infiltrating macrophages are type I interferons. To address whether type I interferon signaling regulates the capacity of monocytes to colonize the brain, we performed a competitive (WT:*Ifnar1*^{-/-}) bone marrow chimera experiment. This experiment demonstrated that *Ifnar1*^{-/-} monocytes had a competitive advantage over WT cells in repopulating the brain, indicating that type I interferons impair colonization of the microglial niche (Fig. R4). This new data is included in a new Fig. 7 in the revised manuscript.

Figure R4: Analysis of chimerism in WT:*Ifnar1*^{-/-} competitive chimeras before TAM administration (blood) or 3 weeks after TAM (CNS).

Further points:

2. CD11b levels appear to be upregulated and remain high in resident microglia even several weeks after repopulation, in contrast to lower levels seen in F4/80^{hi} cells. How does this compare to CD11b levels seen in brain-infiltrating cells before they enter the brain?

This is correct, CD11b levels already become elevated in microglia on day 2 after TAM. Ly6C^{hi} monocytes entering the brain at day 2 express even higher levels of CD11b. However, F4/80^{hi} macrophages express CD11b levels comparable or lower to those in naïve microglia. These data are consistent with the interpretation that Ly6C^{hi} monocytes downregulate CD11b levels upon engrafting the brain (Fig. R5).

Figure R5: Cd11b levels in F4/80^{low} and F4/80^{hi} macrophages on the indicated days after TAM administration.

3. Analysis of Ly6C^{low} monocytes in the blood during depletion (Suppl. Fig. 2F) suggests a reduction of these cells in response to tamoxifen administration. Please explain. Was the actual „0 day” sample taken immediately after the 3rd tamoxifen treatment?

The Ly6C^{low} monocytes are efficiently depleted due to their high expression of CX3CR1. The 0 day time point was taken 24 hours after the second tamoxifen treatment.

4. At d28 there seems to be an increase in CX3CR1- / F4/80- cells in the brain (Fig 1D). Have the authors observed a recruitment of granulocytes, T cells or B cells into the brain after microglia depletion?

In contrast to Ly6C^{hi} monocytes, there is no elevation of neutrophils (CD11b⁺CD45⁺Ly6G⁺) or lymphocytes (CD11b⁻CD45^{hi}) following depletion (Fig. R6).

Figure R6: Total number of neutrophils and lymphocytes in the CNS on the indicated days after TAM administration.

5. Monocyte recruitment is expected to be associated with the upregulation of vascular adhesion molecules – it would be useful to show whether microglia depletion contributed to vascular activation in these studies or the route of cell infiltration into the brain perenchyma was primarily via the meninges / choroid plexus.

This was a great suggestion and we have performed the suggested experiment. We depleted microglia and sacrificed mice on day 2 since this was the time point of maximum Ly6C^{hi} monocyte infiltration. Vascular ICAM-1 staining was significantly elevated at this time (Fig. R7A). This data has been included in Fig. 3D in the revised manuscript. Iba-1 staining in parallel did not reveal obvious infiltrates in the meninges or choroid plexus, arguing against these anatomical areas as major infiltration sites (Fig. R7B).

Figure R7: (A) ICAM-1 staining in brain sections on day 2 after TAM administration. (B) Iba-1 staining in meninges and choroid plexus on day 2.

6. Whilst resident microglia appear to return to a more homeostatic state 12 weeks after depletion (4 weeks after tamoxifen administration 361 differentially expressed genes compared to naive microglia, and 56 genes after 12 weeks), recruited blood-borne macrophages appear to increase their differentially expressed genes over time (886 and 1173 differentially expressed genes, 4 weeks and 12 weeks after depletion,

respectively). This difference is also visible in the heat maps presented in Fig.5D. Do the authors have an explanation to this? Is it possible that unlike in the case of long-lived microglia, recruited macrophages die over time and are replaced by additional waves of blood-borne monocytes several weeks after depletion?

This is an interesting point brought up by the reviewer and in our EdU experiments we have only found support for an early wave of monocyte infiltration. When EdU is administered during a 14 day period, CNS-retrieved Ly6C^{hi} monocytes are consistently ~80% EdU⁺. When EdU was given during days 0-14 after TAM, the F4/80^{hi} monocyte-derived macrophages are also ~80% EdU⁺. This concurs with Fig. 3C in the manuscript, demonstrating that this time point is when monocytes engraft the brain. However, when EdU is administered either during days 14-28 or even later during days 56-80, EdU incorporation into the F4/80^{hi} pool is not different from controls (Fig. R8).

Figure R8: EdU incorporation (14 days in drinking water) in CNS infiltrating monocytes and F4/80hi macrophages when given during the indicated time periods after TAM administration.

Reviewer #2 (immune-neuro crosstalk, NK):

In this work the authors are demonstrating that microglia can be repopulated by peripheral monocytes if there is a niche/room provided through inability of microglia to proliferate and fill the niche. The manuscript is very well done and is addressing an important topic. The results are somewhat novel compared and controversial to decades of previous studies suggesting monocytes cannot repopulate microglia unless the brain is irradiated. However, in the last few months two papers have appeared which are showing very similar findings, which to some extent erode the novelty of this finding. The previous works (Cronk et al., JEM 2018 and Bennett et al Neuron 2018) are not cited or discussed here. Especially the JEM paper is virtually showing very similar things to these findings. It would be interesting if the authors have more closely compared their findings to those made previously. The authors are using fewer models and less advanced bioinformatics than the other two studies, but they get into more nuance by showing that the monocyte-derived macrophages take on some of the properties of microglia, yet ultimately maintain a unique identity. Overall, this is an interesting story and upon minor revision addressing the comments above, it would make a strong contribution to the current literature on microglia.

The reviewer has a very good point, and the reason the Cronk or Bennett papers were not included in our original manuscript is that they were published *after* our initial submission. We have now performed an extensive comparison of our transcriptomic profiles with those published in several previous studies, and this data is included as a new Fig. 6. Firstly, we defined gene signatures of infiltrating monocyte-derived macrophages (F4/80^{hi}) and proliferating microglia (F4/80^{low}) in our model. We then explored the expression of these gene signatures in published datasets comparing CNS infiltrating macrophages to microglia. This included the three models used in the Cronk paper. It also included key populations intracerebrally transplanted into an empty microglial niche, used in the Bennett paper. This analysis demonstrated that our F4/80^{hi} gene signature was highly expressed in CNS infiltrating macrophages in several microglia depletion systems, and that our F4/80^{low} signature was expressed in naive and proliferating microglia as well as intracerebrally transplanted microglia. Finally, we also explored if previously defined gene signatures from CNS infiltrating macrophages vs microglia were enriched in our dataset. This included the beMφ-50 and the Mg-52 gene signatures presented in the Cronk paper, which were strongly enriched in our F4/80^{hi} and F4/80^{low} transcriptomic profiles, respectively (Fig. R9).

Figure R9: (A) Top 50 up/down regulated genes in infiltrating F4/80^{hi} macrophages vs resident proliferating F4/80^{low} microglia. (B) Expression (z-scores) of the F4/80^{low} and F4/80^{hi} gene signatures from published datasets. (C) Enrichment of published gene sets in the F4/80^{low} and F4/80^{hi} transcriptomic signatures using BubbleGum. Color indicates the cell subset showing enrichment, and the size and color of circles represent enrichment score and significance, respectively.

Reviewer #3 (epigenetic):

In my view, these data add little to the paper and its message and in fact to some extent they weaken it technically. At best, if my CNS expert co-reviewers consider this work worth publishing, I would advise relegating the DNA methylation data to the Supplemental Material, toning down the relevant section in the text and making it extremely clear that this analysis is purely exploratory.

The reviewer makes a convincing argument, and is correct in saying that the method is purely exploratory. We have decided to proceed as the reviewer suggests and have moved the data to supplementary figures and have both toned down and reduced the discussion of these data.

REVIEWERS' COMMENTS:

Reviewer #1 (Remarks to the Author):

By performing the type I interferon BM chimeric experiment and the comparative data set analysis including other, recently published models the authors provide further interesting observations that are important for the field. My other questions have been answered appropriately.

Minor point: please correct the typo blood brain „abrier“ in the abstract

Reviewer #3 (Remarks to the Author):

The changes the authors made in response to our previous comments satisfactorily address our concerns.

Reviewer #1 (Remarks to the Author):

By performing the type I interferon BM chimeric experiment and the comparative data set analysis including other, recently published models the authors provide further interesting observations that are important for the field. My other questions have been answered appropriately.

Minor point: please correct the typo blood brain „abrier” in the abstract

This typo has been corrected

Reviewer #3 (Remarks to the Author):

The changes the authors made in response to our previous comments satisfactorily address our concerns.